# Performance of a Support Vector Machine Learning Tool for Diagnosing Diabetic Retinopathy in Clinical Practice

**DOI:** 10.3390/jpm13071128

**Published:** 2023-07-12

**Authors:** Tobias P. H. Nissen, Thomas L. Nørgaard, Katja C. Schielke, Peter Vestergaard, Amar Nikontovic, Malgorzata Dawidowicz, Jakob Grauslund, Henrik Vorum, Kristian Aasbjerg

**Affiliations:** 1Steno Diabetes Center North Jutland, 9000 Aalborg, Denmark; 2Department of Ophthalmology, Aalborg University Hospital, Hobrovej 18, 9000 Aalborg, Denmark; 3Department of Clinical Medicine and Endocrinology, Aalborg University Hospital, 9000 Aalborg, Denmark; 4Department of Ophthalmology, Odense University Hospital, 5000 Odense, Denmark; 5Himmerland Eye Clinic, 9600 Aars, Denmark

**Keywords:** machine learning, diabetic retinopathy, screening, RetinaLyze

## Abstract

Purpose: To examine the real-world performance of a support vector machine learning software (RetinaLyze) in order to identify the possible presence of diabetic retinopathy (DR) in patients with diabetes via software implementation in clinical practice. Methods: 1001 eyes from 1001 patients—one eye per patient—participating in the Danish National Screening Programme were included. Three independent ophthalmologists graded all eyes according to the International Clinical Diabetic Retinopathy Disease Severity Scale with the exact level of disease being determined by majority decision. The software detected DR and no DR and was compared to the ophthalmologists’ gradings. Results: At a clinical chosen threshold, the software showed a sensitivity, specificity, positive predictive value and negative predictive value of 84.9% (95% CI: 81.8–87.9), 89.9% (95% CI: 86.8–92.7), 92.1% (95% CI: 89.7–94.4), and 81.0% (95% CI: 77.2–84.7), respectively, when compared to human grading. The results from the routine screening were 87.0% (95% CI: 84.2–89.7), 85.3% (95% CI: 81.8–88.6), 89.2% (95% CI: 86.3–91.7), and 82.5% (95% CI: 78.5–86.0), respectively. AUC was 93.4%. The reference graders Conger’s Exact Kappa was 0.827. Conclusion: The software performed similarly to routine grading with overlapping confidence intervals, indicating comparable performance between the two groups. The intergrader agreement was satisfactory. However, evaluating the updated software alongside updated clinical procedures is crucial. It is therefore recommended that further clinical testing before implementation of the software as a decision support tool is conducted.

## 1. Introduction

Diabetes is a leading cause of severe visual impairment and blindness throughout the world. The prevalence of patients with diabetes has increased rapidly and was estimated to be 463 million in 2019, and it is estimated that this number will be as high as 783 million in 2045 with the highest percentage (79%) of patients living in low- and middle-income countries [1]. DR is reported to be the single most preventable form of blindness in the working-age (20–74 years) population of the United States alone [2,3]. The need for ophthalmologists and trained technicians required for screening is increasing worldwide, and the current prognosis cannot be met earlier than 2040 [4,5]. Optimisation of the DR screening through the use of trained technicians and telemedicine with reading centres is still both cost- and labour-intensive, mostly because of the heavy draw on human resources. Automated computer-guided decision support tools may reduce the need for skilled labour in this area.

Decision support tools may be implemented by the use of machine learning software implemented through artificial intelligence (AI) in an automated screening where software either fully and independently grades images or partly grades or marks DR changes. Several commercial solutions are already available, and AI software for diagnosing or assisting in the diagnosing of DR is a growing industry. IDx-DR was among the first AI tools to be approved by the U.S. Food and Drug Administration (FDA) to automate the detection of greater than mild DR [6]. Software analysis of retinal images has seen significant progress recently especially after the introduction of capable hardware and algorithms for applying a subtype of AI called deep learning [7]. The performance of deep learning compared to traditional machine learning is superior when applied to images, but it comes with some challenges. Specifically, deep learning algorithms need to be trained on huge, typically non-public datasets with a sufficient variety of ethnic phenotypes.

DR scoring systems and equipment can vary from dataset to dataset which creates inconsistencies between lab performance and real-world performance. The differences in lab vs. real-world performance are due to algorithms not typically being applied to the identical population as the one on which it was trained. The software may perform differently depending on the ethnicity of a person as this correlates greatly to the retinal pigment epithelium [8,9,10], as well as the digital fundus camera and number of retinal photos used [11,12,13], thus possibly causing misinterpretations by the algorithm. Despite these challenges, deep learning systems assessing fundus photos from patients with diabetes have shown high specificity and sensitivity in the laboratory compared to retinal specialists [14] when trained and implemented properly. In real-world performance studies, only a few of the commercially available systems performed well (Algorithm G: sensitivity of 80.47% and specificity of 81.28% on a regraded sub-dataset). This was probably caused by the difficulties previously mentioned as described by Lee et al. [15].

The availability of digital fundus cameras has increased, and it has become more common to take multiple retinal photos [11,12,13] in DR screening to cover a wider area due to early-stage DR changes in the periphery. This is a challenge for the generalisability of DR screening software as it has typically been developed on a dataset with its own characteristics, i.e., camera types and ethnicity, which may introduce bias if applied to another population. Clinical validation of software is therefore important if used in settings other than what the original development and validation intended.

Another subtype of machine learning software is support vector machine learning (SVML), which may be used to determine whether retinopathy is present or absent in an image. First described in 2003 by N Larsen et al. and M Larsen et al. [16,17], this software was developed based on two datasets with 137 patients with 260 non-photocoagulated eyes and 100 patients with 200 eyes. Both datasets used digitalised 35 mm colour transparency film with one 60-degree foveal fundus image per eye. The software has since been updated and was reintroduced to the market in 2013. Few if any studies have evaluated if the updated software can correctly detect the presence or absence of DR in a multi-image screening setting with five fundus images per eye.

The objective of this study was to evaluate the performance of the updated software from RetinaLyze A/S on a reference labelled dataset in order to compare the performance of the routine grading to the reference dataset and to evaluate the inter- and intragrader performance of the reference labelled dataset, as the advantage with SVML is that the analysed results are easily explainable for the clinician as shown below.

## 2. Materials and Methods

Based on a power calculation, this study was performed on a new larger population than the original study’s population. In our study, we deviated from the original studies [16,17] by utilising five images per eye instead of one image per eye. This divergence reflects the change in clinical procedures for DR screening in our current clinical setting where the updated protocol involves capturing five images per eye. We, therefore, found the software needs to be tested again due to more images per eye used today in the clinic and because the software has been updated and is being used commercially in different locations.

The fundus photos used in this study were taken retrospectively from Steno Diabetes Center North Jutland’s (Denmark) DR screening programme database acquired in a hospital setting during 2019–2020. A total of 1001 patients from the period were included. Each is represented with one eye. The vast majority of participants were Caucasian. Other ocular comorbidities were not registered. A power calculation using McNemar’s test was performed by two statisticians and showed a minimum sample size of 960 eyes with a power of 90% to detect DR and a delta difference of 5%. Inclusion criteria were a history of any form of diabetes with an ICD-10 DE11*–DE14* diagnosis and a total of five photos per eye—i.e., one fovea-centred image, one papillary image and three peripheral images—as per hospital standards for screening. Patients with previous panretinal laser-treated eyes were included in this study. In the following, we describe the dataset.

After obtaining the photos, routine grading was performed on the images by doctors. For the purpose of this study, additional steps were taken. First, the software assessed the photos, and then, three certified retinal specialists performed the “gold standard” reference grading. Finally, the results from routine, software and reference grading were compared statistically.

More information regarding the photographic technique and software is found in Appendix A.

### 2.1. Routine Grading

The eyes were originally graded from 2019 to 2020 by a mix of 10 ophthalmologist consultants and senior registrars who had varying experiences in clinical care. The International Clinical Diabetic Retinopathy Severity Scale (ICDR) [18] was used for grading. The scale is derived from the ETDRS study and is one of the most commonly used scales for grading DR [19,20]. Routine grading was performed prior to and without knowledge of the study.

### 2.2. Reference Grading (“Gold Standard”)

All 1001 eyes were reference graded independently using the free academic version of Labelbox [21] by three experienced ophthalmologist consultants from two different eye departments whose grading was used as a golden reference standard. The eyes were shown in random order, and each ophthalmologist was allowed as much time as needed for grading. All three ophthalmologists had passed a grading course prior to the study to ensure consensus and uniformity [22]. The five-stage International Clinical Diabetic Retinopathy Disease Severity Scale [20] was used for grading, and the grading was done on all 5 images available per eye. Each eye was scored according to ICDR. Each eye had one fovea-centred image, one papillary image and three peripheral images. The final grade for each eye was determined by majority vote. If there was disagreement between all three graders, the eye was discarded from the analysis.

### 2.3. Automated Grading

The images were analysed by commercially available software from RetinaLyze A/S [23] (henceforth, “the software”). The technology behind the software has been described thoroughly by Larsen et al. [17]. No performance studies of the software have been performed since the reintroduction of the software to the market in 2013 other than a pilot study [24,25]. The software has been slightly updated since its reintroduction with improved analysation time and the ability to grade an image as non-gradable.

The software marks the analysed red lesions, as shown on the central fundus image Figure 1b, compared to the original fundus image Figure 1a. It does not detect all red lesions as seen at the paramacular inferior bleeding. Each marking counts as one red lesion detected on an image. All the red lesions on one image are summed to a single numeric value and summed with the numeric values from the other four images from one eye. The software is only capable of detecting red lesions and not other characteristics of DR such as hard exudates, cottonwool spots or neovascularisations. Further technical explanation is provided in Appendix A.

### 2.4. Statistical Analysis

Sensitivity, specificity, positive predictive value, negative predictive value and accuracy were calculated for routine grades vs. reference grades and software grades vs. reference grades. The threshold for the software was chosen according to an individual and a combined image grading strategy described more thoroughly in Appendix A. The most clinically relevant results are included in this paper. The rest can be found in the Appendix A.

As three graders had graded the entire dataset, the intergrader and intragrader variability was calculated using Conger’s Exact Kappa for multiple graders and intraclass correlation coefficients (ICC) type 2 for intragrader variability.

## 3. Results

Of the 1001 screened eyes, a total of 11 eyes were excluded due to the lack of a majority decision among the retinal specialists. The retinal specialists further rated 10 eyes as ungradable due to cataracts, asteroid hyalosis and insufficient image coverage of the retina.

The software described in this paper was developed for diagnosing any presence of DR and not for DR requiring treatment. The software rated 19 eyes as ungradable. The eye with asteroid hyalosis was rated as having no DR and not as ungradable by the software. One eye with cataract blur was rated ungradable by the software but not by retinal experts.

Six eyes were rated as ungradable by both the specialists and the software. One eye with good image quality, but with insufficient coverage of the retina, was excluded by the retinal specialists. A total of 34 eyes were excluded.

A total set of 967 eyes–509 right eyes and 458 left eyes–from 967 patients were included for further analysis. Of the included patients, 730 patients had type 1 diabetes, 230 had type 2 diabetes, 9 had gestational diabetes and 29 had ICD-10 [26] diagnosis E13.* (Other specified diabetes mellitus) and E14.* (Unspecified diabetes mellitus).

The average grading time for each ophthalmologist was 44 s, 55 s and 77 s, respectively, with a total average of 58 s per eye.

### 3.1. Results at Low Threshold

The results with a threshold with ≥1 red lesion/eye from the software were compared to the reference graded dataset with 967 included eyes and are included in Table 1a. As the threshold of 1 revealed identical results regardless of individual or combined scoring, the numbers are only represented once.

Of the 18 eyes rated as false negatives by the software, seven were graded by the retinal specialists as mild DR, nine as moderate DR, one as severe DR with a large papillary confined haemorrhage and one as proliferative DR (active).

The result was achieved with the standard settings of the software. The area under the curve (AUC) was 93.4% with the ROC curve seen in Figure 2. The software correctly identified 96.8% (95% CI: 95.3–98.2) of the patients with DR and 51.7% (95% CI: 46.8–56.6) without DR. The overall accuracy of the software was 78.0% (95% CI: 75.5–80.6). The positive predictive value (PPV) was 73.6% (95% CI: 70.5–76.8) and the negative predictive value (NPV) was 92.1% (95% CI: 88.3–95.4) (Figure 3).

### 3.2. Reference Grading vs. Routine Grading

The reference graded eyes were compared to the gradings acquired through daily DR routine screening. This showed sensitivity at 87.0% (95% CI: 84.2–89.7), specificity at 85.3% (95% CI: 81.8–88.6), positive predictive value at 89.2% (95% CI: 86.3–91.7) and negative predictive value of 82.5% (95% CI: 78.5–86.0) with an accuracy of 86.3% (95% CI: 84.1–88.4). A total of 964 reference-graded eyes were used to compare the routine gradings of the 964 eyes. For three eyes, routine grades were not available. See Table 1b and Figure 3.

### 3.3. Results with Higher Thresholds

For individual image grading with a threshold ≥ 2 red lesions per eye, a sensitivity of 82.6% (95% CI: 79.4–85.6), specificity of 87.1% (95% CI: 83.8–90.3), PPV of 89.9% (95% CI: 87.3–92.4) and NPV of 78.2% (95% CI: 74.3–81.9) was shown. The accuracy was 84.5% (95% CI: 82.2–86.8). With a threshold ≥ 3 red lesions per eye, sensitivity dropped to 72.6% (95% CI: 68.9–76.3), but specificity went up to 95.3% (95% CI: 93.2–97.2). PPV and NPV were 95.6% (95% CI: 93.5–97.4) and 71.4% (95% CI: 67.6–75.2), respectively, with an accuracy of 82.1% (95% CI: 79.7–84.5). The results are shown in Table 1c,d and Figure 3.

For combined image grading, a threshold ≥ 2 red lesions per eye showed a sensitivity of 90.8% (95% CI: 88.3–93.1), specificity of 77.5% (95% CI: 73.3–81.5), PPV of 84.9% (95% CI: 81.9–87.7), NPV of 85.8% (95% CI: 82.0–89.3) and accuracy of 85.2% (95% CI: 82.9–87.4). For a threshold of ≥3 red lesions, this setting resulted in slightly lower sensitivity at 84.9% (95% CI: 81.8–87.9), a specificity of 89.9% (95% CI: 86.8–92.7), PPV of 92.1% (95% CI: 89.7–94.4) and NPV of 81.0% (95% CI: 77.2–84.7) with higher accuracy at 87.0% (95% CI: 84.8–89.0). Please see Table 1d,f and Figure 3.

### 3.4. Grader Variability

For the intergrader variability correlation between the three reference graders’ gradings, the variability was calculated using Conger’s Exact Kappa [27] method for multiple graders. The Kappa (Κ) value ranged from 0 (no agreement) to 1 (perfect agreement). Values between 0.8≥ Κ ≥0.61 show substantial agreement. Κ > 0.8 is almost perfect agreement [28]. For the ICDR grading, this resulted in a Kappa value of 0.731. The binary classification of the ICDR class 0 was defined as no DR and classes 1–4 were defined as DR. This resulted in a Kappa of 0.827.

Intragrader (X compared to Y) variability was available for two of the three reference graders. The two reference graders had previously screened 59 and 132, respectively, of the included eyes as part of the routine grading. The intraclass correlation coefficient (ICC2 (2,1)) was calculated for both graders. Grader Y ICC: 0.81 (95% CI: 0.72–0.88). Grader X ICC: 0.90 (95% CI: 0.86–0.92). ICC between 0.75 and 0.9 indicates good reliability, and ICC greater than 0.90 indicates excellent reliability [29].

## 4. Discussion

In this study, we demonstrated the ability of red lesion detection software to detect the presence or absence of DR in a five-field fundus photo screening. We applied two different strategies. (1) Individual image grading where each of the five images’ red lesion scores were set individually, and (2) combined image grading where the red lesion scores were summed for all five images. A red lesion threshold was then set to assess how software performance changes as the threshold changes according to the individual and combined image grading strategy. At the software base settings, only one red lesion in one of the five fundus photos per eye with a high sensitivity of 96.8% (CI 95%: 95.3–98.2) but low specificity of 51.7% (CI 95%: 46.8–56.6) was shown to have the same result for both the individual and combined image grading. We utilised a single red lesion as a threshold that affects the specificity and, thus, leads to a higher number of false positives. At a threshold of two red lesions, the two strategies diverged. The individual image grading showed better-weighted performance with higher sensitivity and specificity, but the combined image grading strategy dropped only slightly in sensitivity but increased in specificity (Figure 3).

The most ideal approach would yield results that are comparable to the grading performed in the clinical routine. By using individual image grading as a strategy, we lost information for each ascending threshold. Our observations showed the best-balanced performance of the individual image grading was a threshold of a minimum of ≥2 red lesions per eye for categorising the eye as having DR (Figure 3). This strategy of reporting is inferior to the combined image grading strategy and should preferably be avoided as individual image grading strategy results are not all included in or better than the confidence intervals of routine grading as shown in Figure 3.

In the combined image grading strategy, with a threshold of a minimum of ≥2 red lesions per eye for categorising the eye as having DR, the results were more comparable to the results from routine grading. The software had a better NPV compared to the routine grading 85.8% (CI 95%: 82.0–89.3) vs. 82.5% (CI 95%: 78.5–86.0), which could be important if the software assists in diagnosing DR. At a threshold of ≥ 3 red lesions per eye, sensitivity was a bit lower than the sensitivity of routine grading, but specificity was greatly improved from 77.5% (95% CI: 73.3–81.5) to 89.9% (95% CI: 86.8–92.7) compared to a threshold of ≥2 red lesions per eye, both of which were compared to the routine grading and resulted in fewer false positives.

As seen in Figure 3, the confidence intervals of the combined image grading strategy with a threshold ≥ 3 red lesions per eye either overlapped or were superior to routine grading in all of the five categories perhaps making this the best approach for clinical practice. With an increased threshold to three red lesions per eye, the software operated at higher specificity compared to the reference grading. As there was no unambiguous difference in the confidence intervals, we cannot deny the values were the same or that there is a significant difference between the combined image grading strategy with a threshold ≥ 3 red lesions and routine grading. Figure 3 highlights a trade-off between specificity and sensitivity or PPV and NPV as the threshold for red lesions increases.

AUC was 93.4% (Figure 2) compared to reference grading which is decent and comparable to the AUC of the studies by M Larsen et al. and N Larsen et al. [16,17], who reported an AUC of 94.1%.

Compared to the original studies of the software from 2003 [16,17], the software achieved similar specificities, sensitivities and accuracies on a single central fundus photo. This study is, however, not directly comparable to the ones from 2003 by M Larsen et al. and N Larsen et al. because we used five fundus photos per eye and N Larsen and M Larsen et al. used one central fundus photo per eye. The disadvantage of using five fundus photos is an increased chance of false positives increases with the number of photos taken and analysed by a red lesion detection tool due to the overlap of the images and the possibility of a red lesion being counted twice by the software. Stitching the images together was also considered, but this had its own concerns, i.e., decreased image quality in the periphery and the fact that stitching can potentially cover areas that may not be analysed. Using five images per eye may also be an advantage as minor DR changes can show themselves in the periphery. These changes may not be observed if only a central fundus photo is recorded.

Compared to deep learning software, few larger comparison studies have been made to the best of our knowledge. Software tends to perform a bit under lab performance when evaluated on real-world data. In the study by Lee et al. [15], the best-performing software of seven commercially available software included showed a sensitivity of 80.47% and specificity of 81.28%. The software was anonymised. This paper evaluated referrable DR where the threshold for referrable DR may vary from country to country. Lee et al. reported generally high negative predictive values (82.72–93.69%) and a large spread in sensitivity of 50.98–85.90% [15]. A direct comparison of the seven software included in the study by Lee et al. [15] is not feasible, as the SVML in the study does not use referrable DR as a threshold but only categorises whether DR is present or not. The SVML software included in this study performs decently with a sensitivity of 84.9%, specificity of 89.9% and NPV of 81.0% (see Figure 3; combined red lesion threshold: 3).

A strength of this study is that three independent ophthalmologists reviewed the dataset and made a majority decision on the ICDR grade. Furthermore, as the software was not developed on the fundus camera used for making the dataset, the results may be more generalisable to other fundus cameras as well. The explainability of the results is considered good as the software outputs an image with black lines around the red lesions (Figure 1b) which can easily be compared to the original photo (Figure 1a). This is an advantageous form of reporting as both the clinician and patient can easily understand what the software detects and why it scores as it does. Additionally, the software rated only 19 (1.9%) eyes as ungradable which is considered acceptable.

A major limitation of the software is that it lacks the ability to grade according to the ICDR scale. This is important to note as newer deep learning software has been able to accomplish this. Software that is supposed to be used for screening DR should be able to distinguish DR according to the ICDR classification. Another limitation of this study is the dataset’s size compared to other big data studies, and the dataset’s heterogeneity with Caucasians as the primary ethnic group due to the heterogeneity of the population of Denmark.

The software has not been tested for performance on multiple photos per eye before this study. We argue that the tested software can be useful in a screening setting to sort between eyes with disease and without disease more easily or to replace specially trained personnel doing the coarse sorting at screening centres. The red lesion threshold for diagnosing DR should be determined according to local requirements. The software has currently mainly been tested on a primarily Caucasian population and generalisability to other ethnicities is unknown.

The limits of the software are its ability to detect papillary haemorrhages of both DR and moderate DR and not being able to grade the stages of the ICDR scale or make a cut-off for referable DR. To perform at its best, the software should ideally make a collective grade on all the five images per eye as shown in the combined image grading strategy.

The intergrader agreement was comparable to the literature [30,31] with a Conger’s Exact Kappa at 0.731 (95% CI: 0.705–0.757) for three graders at the ICDR scale and at 0.827 (95% CI: 0.798–0.856) at binary grading. For the intragrader variability, the Kappa was calculated using ICC type 2 and showed ICC at 0.81 (95% CI: 0.72–0.88) for grader Y and ICC of 0.90 (95% CI: 0.86–0.92) for grader X which are considered decent scores [29].

## 5. Conclusions

The software exhibited similar performance to the original studies and demonstrated comparability to routine grading. Acceptable levels of intergrader and intragrader variability were observed. However, it should be noted that the software lacks the capability to grade according to the International Clinical Diabetic Retinopathy severity scale. For the software to be implemented as a screening tool, conducting local clinical validation and establishing regular quality control measures is crucial. The accuracy of software-generated reports should be carefully examined as indicated by the performance differences observed in both individual and combined image grading strategies. With the increasing prevalence and incidence of diabetes worldwide, there is a growing need for diabetic retinopathy screening to preserve visual health. Therefore, it is important to prioritise local ophthalmic resources for individuals most in need. Despite the limitations of software analysis, the progress and implementation of DR software analysis can be valuable. We acknowledge the development of deep learning software that offers higher AUC, specificities, and sensitivities is needed. Nevertheless, it is crucial to thoroughly test all software tools before their clinical implementation. In our opinion, clinicians may find it easier to interpret a mark around a red lesion compared to a heat map generated by deep learning software. Considering the SVML software used in this study, conducting this performance study was necessary for evaluating its clinical relevance.

## Figures and Tables

**Figure 1 jpm-13-01128-f001:**
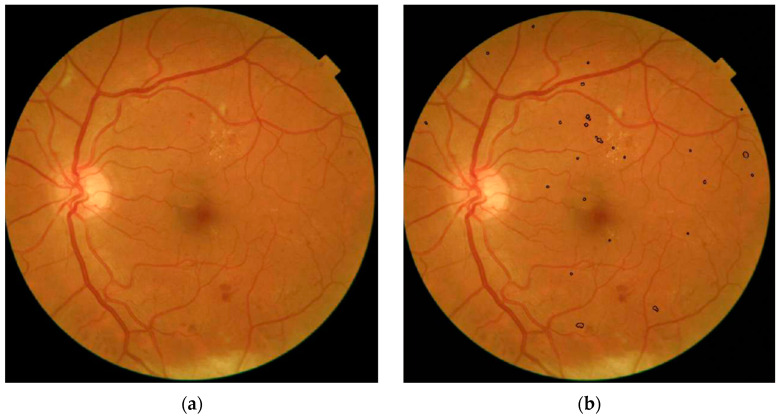
(**a**) No software interpretation present. (**b**) Software interpretation of red lesions marked with black circles.

**Figure 2 jpm-13-01128-f002:**
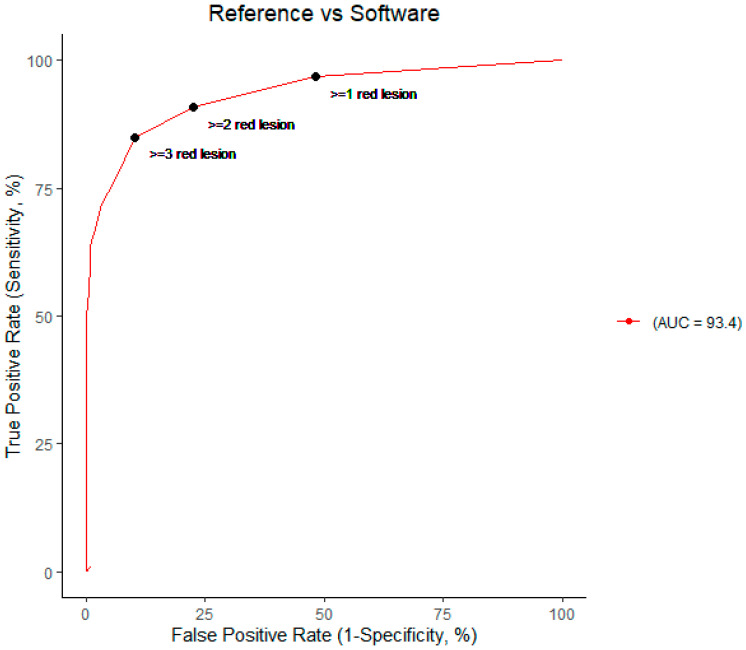
Performance of the software vs. the reference labelled dataset illustrated on a receiver operating curve (ROC) (red line) with an area under the curve (AUC) at 93.4%. The three different thresholds for diagnosing DR according to the software are shown by the three filled circles. The combined image grading strategy is used for this ROC as the individual image grading approach would make a different ROC with a lower AUC for each increase in the red lesion threshold.

**Figure 3 jpm-13-01128-f003:**
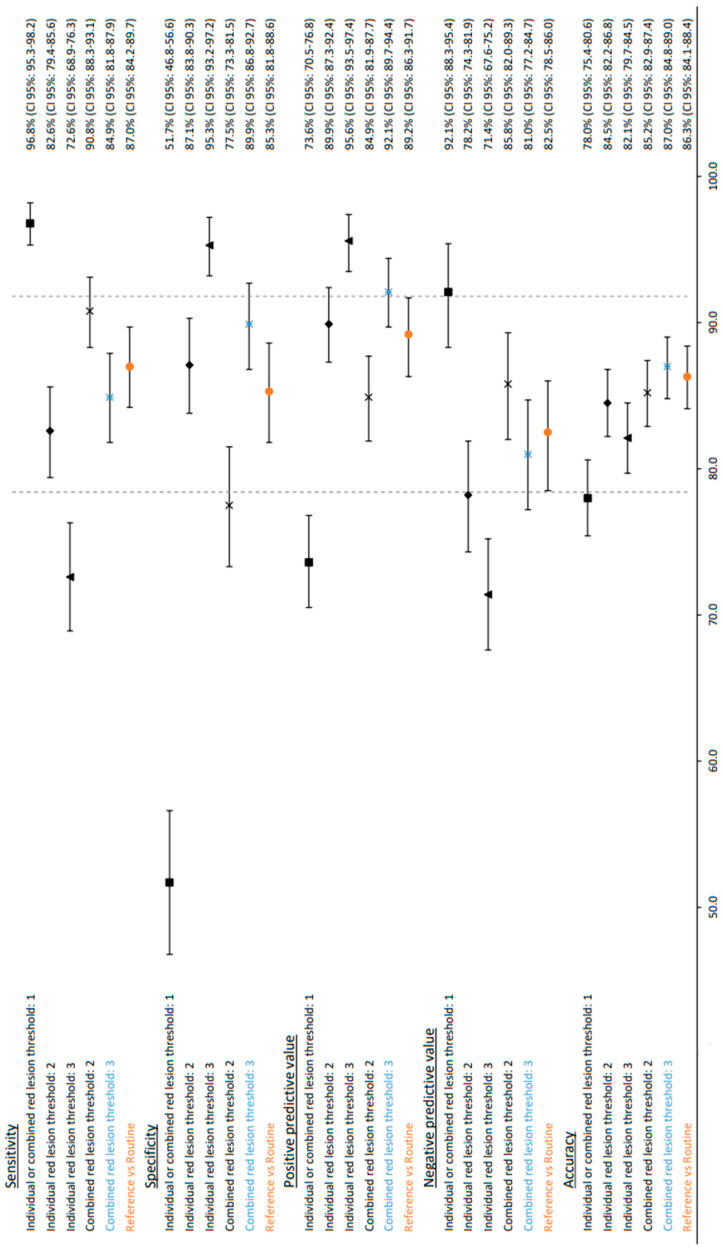
Forest plot of the different sensitivities, specificities, positive predictive values, negative predictive values and accuracy. *X*-axis is a percentage. Individual image grading and combined image grading are carried out as described in the methodology. The dotted vertical lines represent the outermost CI for the results of routine screening (orange). The best comparable threshold and strategy is the combined image grading with a threshold of three red dots (blue).

**Table 1 jpm-13-01128-t001:** Classification of patients with diabetes by absence or presence of diabetic retinopathy on fundus images.

	Reference				Reference		
Software	DR	No DR	Total	Routine	DR	No DR	Total
**DR**	545	195	740	**DR**	489	59	548
**No DR**	18	209	227	**No DR**	73	343	416
**Total**	563	404	967	**Total**	562	402	964
(**a**) Individual and combined grading. Error matrix of the reference grading vs. automated grading by the software ≥ 1 red lesion per eye. Number of eyes.				(**b**) Error matrix of the reference grading vs. routine grading. Number of eyes.			
	**Reference**				**Reference**		
**Software**	**DR**	**No DR**	**Total**	**Software**	**DR**	**No DR**	**Total**
**DR**	465	52	517	**DR**	511	91	602
**No DR**	98	352	450	**No DR**	52	313	365
**Total**	563	404	967	**Total**	563	404	967
(**c**) Individual image grading. Error matrix of the reference grading vs. automated grading by the software ≥ 2 red lesions per eye. Number of eyes.				(**d**) Combined image grading. Error matrix of the reference grading vs. automated grading by the software ≥ 2 red lesions per eye. Number of eyes.			
	**Reference**				**Reference**		
**Software**	**DR**	**No DR**	**Total**	**Software**	**DR**	**No DR**	**Total**
**DR**	384	18	402	**DR**	478	41	519
**No DR**	179	386	565	**No DR**	85	363	448
**Total**	563	404	967	**Total**	563	404	967
(**e**) Individual image grading. Error matrix of the reference grading vs. automated grading by the software ≥ 3 red lesions per eye. Number of eyes.				(**f**) Combined image grading. Error matrix of the reference grading vs. automated grading by the software ≥ 3 red lesions per eye. Number of eyes.			

## Data Availability

Data is not available due to restrictions regarding anonymity.

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
