# Peer review of "Performance of a Support Vector Machine Learning Tool for Diagnosing Diabetic Retinopathy in Clinical Practice"

_jpm, 2023, doi:10.3390/jpm13071128_

Round 1

Reviewer 1 Report

It is a good article on use of SVML for detection of diabetic retinopathy

1. Introduction is very long, can be shortened

2. When we speak about ICDR severity scale used in the study, it refers to 5 grades of DR, whereas this algorithm only detect red lesions, which is one of the many DR lesions, it doesn't provide details of grades of DR based on ICDR or any information on referable DR. It is one of the major limitations of this study.

3. The specificity of the algorithm appears to be lesser. 

4. Now there are many AI algorithms  that provide sensitivity and specificity beyond 90-95%. When using 5 images per eye, the accuracy of the algorithm should be even higher

5. Kindly provide more details/ references comparing this with other AI based DR detection tools in the discussion.

6. Was the same fundus camera used for all patients? What was the field of view... were the retinal photographs taken with or without mydriasis? These details can be added to the methods section

7. How many images were ungradable by the algorithm and how many were ungradable by the 3 human graders?

8. What was the time taken by the algorithm to process results for one patient?

No specific comments. 

Author Response

Dear Reviewer

Thank you for your comments. We have addressed your comments point by point in the following. 

  1. Thank you for your comment. We have tried to shorten the introduction. We hope you find this sufficient. We find it important to contextualize why we test SVML software when deep learning is surely on the rise.  
  2. Thank you for noticing this. We have added this to the Discussion section under limitations at lines 348-350. 
  3. Thank you for your comment. As we understand your comment, it is a more general comment that we agree upon.
  4. We agree with your comment that the sensitivity and specificity are below those of deep learning systems, we refer to a study by Lee et al. (references 15), where a direct comparison of seven commercial software is made. 
  5. We have updated the discussion with a comparison to other real-world tests of DR software. Please see line 325-337 
  6. Thank you for the comment. We are sorry we were unclear, and we have made a few changes to this in Appendix A.1 section where we describe that we use mydriasis, the same model Topcon camera at four different locations. Please see lines 409-414.
  7. Thank you for the comment. We are sorry we were unclear, and we have made a few changes in section 3. Results, lines 168-178 to clarify this. 
  8. Thank you for your comment, we have added this information to Appendix A.1 lines 414-416. We hope you find this information sufficient. 

We hope we have understood and answered your valuable comments satisfactorily.

Best regards

Reviewer 2 Report

Dear Authors,

Performance of a support vector machine learning tool for diagnosing diabetic retinopathy in clinical practice

This is an interesting article addressing a topic that is increasingly addressed in medical imaging analysis. Software must be more and more efficient, safe and flexible to adapt to different imaging devices. This requires the publication of articles like this one to validate the last updates.

There is a good explanation of the protocol and methodology.

Minor comments: 

-             There are sometimes too many repetitions making reading less comfortable (line 94 to 97; line 109 and 135, figure and text…)

Best regards

Author Response

Dear Reviewer

Thank you for your review, and pointing out the repetitions.

We have edited your highlights and looked through the text to make it less repetitive. We hope you find these changes sufficient. 

Best regards

Round 2

Reviewer 1 Report

Inability to provide ICDR classification DR grading is a major limitation.

Just detection red lesions of DR is insufficient for a software. What about hard exudates, cottonwool spots, new vessels, etc? 

in line 353, what does papillar signs of DR mean?

Author Response

Dear reviewer

Thank you for the time to edit our paper. You find our response down below.

1) Thank you for your comment. We have clarified that not being able to classify according to ICDR is a major limitation. Please see lines 349-352.

2) Thank you for your comment. We have clarified that the software is only able to find red lesions. Please see lines 149-151, and repeated this in the Appendix lines 451-452

3) Thank you for your comment. We see it wasn't as clearly expressed as intended. In line 353 (now 362) we have edited the language to make it more clear. It was a papillary haemorrhage the software did not detect, and 

Additionally, we have had the language revised by a professional British native-speaking academic English specialist. We hope you find the result satisfactory.

We you find our answers satisfactory.  

Best regards

Round 3

Reviewer 1 Report

The conclusion in the abstract and the main text can be made more precise pertaining to the article alone.

Rest of the edits in the manuscript done are satisfactory

Kindly check grammar . The tense is changing  throughout the manuscript. Kindly maintain uniformity .

Author Response

Thank you for the comments. 

1) The conclusions in the abstract and main text have been rewritten.

2) The paper has been through another professional English revision with a special focus on the tense. 

We hope you find the changes satisfactory.